# Analogous to Evolutionary Algorithm: Designing a Unified Sequence Model

**Jiangning Zhang**[1][*] **Chao Xu**[1] **Jian Li**[2] **Wenzhou Chen**[1] **Yabiao Wang**[2]
**Ying Tai**[2] **Shuo Chen**[3] **Chengjie Wang**[2] **Feiyue Huang**[2] **Yong Liu**[1][†]

[1]APRIL Lab, Zhejiang University  [2]Youtu Lab, Tencent
[3]RIKEN Center for Advanced Intelligence Project

`{186368, 21832066, wenzhouchen}@zju.edu.cn, yongliu@iipc.zju.edu.cn`
`{swordli, caseywang, yingtai, jasoncjwang, feiyuehuang}@tencent.com`
`shuo.chen.ya@riken.jp`

## Abstract

Inspired by biological evolution, we explain the rationality of Vision Transformer by analogy with the proven practical Evolutionary Algorithm (EA) and derive that both of them have consistent mathematical representation. Analogous to the dynamic local population in EA, we improve the existing transformer structure and propose a more efficient EAT model, and design task-related heads to deal with different tasks more flexibly. Moreover, we introduce the spatial-filling curve into the current vision transformer to sequence image data into a uniform sequential format. Thus we can design a unified EAT framework to address multi-modal tasks, separating the network architecture from the data format adaptation. Our approach achieves state-of-the-art results on the ImageNet classification task compared with recent vision transformer works while having smaller parameters and greater throughput. We further conduct multi-modal tasks to demonstrate the superiority of the unified EAT, *e.g.*, Text-Based Image Retrieval, and our approach improves the rank-1 by +3.7 points over the baseline on the CSS dataset.[3]

## 1 Introduction

Since Vaswani *et al.* [69] introduce the Transformer that achieves outstanding success in the machine translation task, many improvements have been made to this method [15, 38, 21]. Recent works [64, 82, 43] led by ViT [23] have achieved great success in the field of many vision tasks by replacing CNN with transformer structure. In general, these works are experimentally conducted to verify the effectiveness of modules or improvements, but they may lack other forms of supporting evidence.

Inspired by biological population evolution, we explain the rationality of Vision Transformer by analogy with the proven effective, stable, and robust Evolutionary Algorithm (EA), which has been widely used in practical applications. Through analogical analysis, we observe that the training procedure of the transformer has similar attributes to the naive EA, as shown in Figure 1. Take the one-tier transformer (*abbr.*, TR) as an example. *1)* TR processes a sequence of patch embeddings while EA evolutes individuals, both of which have the same vector formats and necessary initialization. *2)* The Multi-head Self-Attention (MSA) among patch embeddings in TR is compared with that of (sparse) global individual crossover among all individuals in EA, in which local and dynamic population concepts are introduced to increase running speed and optimize results [63, 59]. *3)* Feed-Forward

---

[*]Work done during an intership at Tencent Youtu Lab.
[†]Corresponding author.
[3]Code is available at `https://github.com/TencentYoutuResearch/BaseArchitecture-EAT`.

35th Conference on Neural Information Processing Systems (NeurIPS 2021).

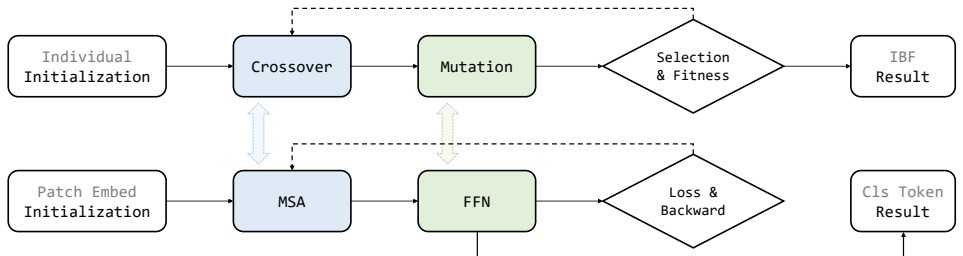

Figure 1: Analogy of EA (top) and Transformer (bottom) pipelines. For simplicity, only one layer of the Transformer structure is displayed here.

Figure 2: Different SFC indexing methods, taking 2D images with side length of 8 as an example.

Network (FFN) in TR enhances embedding features that is similar to the individual mutation in EA. *4)* During training, TR optimizes the network through backpropagation while EA optimizes individuals through selection and fitness calculation. *5)* TR chooses the enhanced Classification Token (Cls Token) as the target output, while EA chooses Individual with the Best Fitness (IBF). Meanwhile, we deduce the mathematical characterization of crossover and mutation operators in EA (*c.f.*, Equations 5,8) and find that they have the same mathematical representation as MSA and FFN in TR (*c.f.*, Equations 6,9), respectively. Inspired by the characteristics in the crossover step of EA, we propose a novel *EA-based Transformer* (EAT) that intuitively designs a local operator in parallel with global MSA operation, in which the local operator can be instantiated as 1D convolution, local MSA, *etc*. Subsequent experiments demonstrate the effectiveness of this design in that it could reduce parameters and improve the running speed and boost the network performance, which is consistent with the results of EAs in turn. Current TR-based models would initialize different tokens for different tasks, and they participate in every level of calculation that is somewhat incompatible with other tokens for internal operations. Therefore, we design task-related heads docked with transformer backbone to complete final information fusion, which is more flexible for different tasks learning and suitable for the transfer learning of downstream tasks.

Our designed local operator receives the same data format as the global MSA branch, *i.e.*, a sequence that is conform to NLP. Therefore, the generally used high dimension operations such as 2D reshape [79, 77, 80] are not required, which brings the possibility to standardize multi-modal data (*e.g.*, 1D sentence, 2D image, and 3D video) of input into consistent sequence data in *one* unified model. To accomplish this target, we introduce the space-filling curve (SFC) concept to standardize the multi-modal data format and design an SFC module. As shown in Figure 2, taking 2D image as an example, the top half represents four kinds of SFCs: Sweep, Scan, Z-Order, and Hilbert, while the bottom two lines represent the sequenced image by Z-Order SFC. Thus the image ($\in \mathbb{R}^{8 \times 8 \times 3}$) is specifically re-indexed and arranged ($\in \mathbb{R}^{64 \times 3}$) by the predefined SFC before feeding the network, realizing uniform sequence input. The difference between SFCs mainly reflects how the 1D sequence preserves the 2D spatial structure, and the SFC can be extended to 3D and higher dimensions.

Specifically, we make the following four contributions:

- In theory, we explain the rationality of Vision Transformer (TR) by analogy with Evolutionary Algorithm (EA) and derive that they have consistent mathematical representation.

- For the method, we improve a unified EAT model by analogy with dynamic local population concept in EA and design a Task-related Head to deal with various tasks more flexibly.

- On framework, we introduce Space-Filling Curve (SFC) module as a bridge between rasterized 2D image data and serialized 1D sequence data. This makes it possible to integrate the unified paradigm that uses a unified model to solve multi-modal tasks, keeping the network architecture and data structure independent of the data dimensionality. Note that only several 1D operators are required for our method, meaning that it is very friendly to the underlying optimization workload of different platforms.

- Massive experiments on classification and multi-modal tasks demonstrate the superiority and flexibility of our approach.

## 2 Related Work

### 2.1 Evolution Algorithms

Evolution algorithm is a subset of evolutionary computation in computational intelligence, and it belongs to modern heuristics [60, 70]. Inspired by biological evolution, general EA contains reproduction, crossover, mutation, and selection steps, and has been proven to be effective and stable in many application scenarios [29]. Moscato *et al.* [47] propose the Memetic Algorithm (MA) for the first time in 1989, which applies a local search process to refine solutions. Later Differential Evolution (DE) appears in 1995 [62] is arguably one of the most competitive improved variant [18, 49, 52, 24]. The core of DE is a differential mutation operator, which differentiates and scales two individuals in the same population and interacts with the third individual to generate a new individual. In contrast to the aforementioned global optimization, Local Search Procedures (LSPs) aim to find a solution that is as good as or better than all other solutions in its "neighborhood" [44, 14, 9, 41, 28]. LSP is more efficient than global search in that a solution can be verified as a local optimum quickly without associating the global search space. Also, some works [44, 14, 41] apply the Multi-Population Evolutionary Algorithm (MPEA) to solve the constrained function optimization problems relatively efficiently. Recently, Li *et al.* [41] point out that the ecological and evolutionary history of each population is unique, and the capacity of Xylella fastidiosa varies among subspecies and potentially among populations. Therefore, we argue that the local population as a complement to the global population can speed up the EA convergence and obtain better results. In this paper, we explain and improve the naive transformer structure by analogy with the validated evolutionary algorithms, where a parallel local path is designed inspired by the local population concept in EA.

### 2.2 Vision Transformers

Since Transformer structure is proposed for the machine translation task [69], many improved language models [51, 21, 53, 54, 7] follow it and achieve great results. Some later works [32, 74, 15, 38, 3] improve the basic transformer structure for better efficiency. Inspired by the high performance of transformer in NLP and benefitted from abundant computation, recent ViT [23] introduces the transformer to vision classification for the first time and sparks a new wave of excitement for many vision tasks, *e.g.*, detection [8, 85], segmentation [81, 11, 66], generative adversarial network [36, 72], low-level tasks [10], video processing [4], general backbone [76, 75, 43], self-supervision [12, 2, 13, 48], neural architecture search [73, 40], *etc*. Many researchers have focused on improving the basic transformer structure [64, 82, 77, 26, 65, 79, 19, 80, 16, 68, 61, 17, 42], which is more challenging than other application-oriented works. Among these methods, DeiT [64] is undoubtedly a star job that makes the transformer performance better and training more data-efficient. Based on DeiT, this work focuses on improving the basic structure of the vision transformer, making it perform better considering both accuracy and efficiency. Besides, our approach supports multi-modal tasks using only one unified model while other transformer-based methods such as DeiT do not.

### 2.3 Space Filling Curve

A space-filling curve maps the multi-dimensional space into the 1D space, and we mainly deal with 2D SFCs in this paper. SFC acts as a thread that passes through all the points in the space while visiting each point only once, *i.e.*, every point except the starting and ending points is connected to two adjacent line segments. There are numerous kinds of SFCs, and the difference among them is in their ways of mapping to the 1D space. Peano [50] first introduces a mapping from the unit interval to the unit square in 1890. Hilbert [33] generalizes the idea to a mapping of the whole space. G.M.Morton [46] proposes Z-order for file sequencing of a static two-dimensional geographical database. Subsequently, many SFCs are proposed [45, 67, 56], *e.g.*, Sweep, Scan, Sierpiński, Lebesgue, Schoenberg, *etc*. Some researchers further apply SFC methods to practical applications [5, 83, 6], such as data mining and bandwidth reduction, but so far, almost none has been

applied to computer vision. By introducing SFC in our EAT, this paper aims to provide a systematic and scalable paradigm that uses a unified model to deal with multi-modal data, keeping the network architecture and data structure independent of the data dimensionality.

## 3   Preliminary Transformer

The Transformer structure in vision tasks usually refers to the encoder and mainly builds upon the MSA layer, along with FFN, Layer Normalization (LN), and Residual Connection (RC) operations.

**MSA** is equivalent to the fusion of several SA operations that jointly attend to information from different representation subspaces, formulated as:

$$
\begin{aligned}
\text{MultiHead}(Q, K, V) &= \text{ Concat ( Head }_1, \text{ Head }_2, \ldots, \text{ Head }_h) W^O, \\
\text{here Head }_i &= \text{Attention}\left(QW_i^Q, KW_i^K, VW_i^V\right) \\
&= \text{softmax}\left[\frac{QW_i^Q\left(KW_i^K\right)^T}{\sqrt{d_k}}\right] VW_i^V = AVW_i^V,
\end{aligned}
\tag{1}
$$

where $W_i^Q \in \mathbb{R}^{d_m \times d_k}, W_i^K \in \mathbb{R}^{d_m \times d_k}, W_i^V \in \mathbb{R}^{d_m \times d_v}, W^O \in \mathbb{R}^{hd_v \times d_m}$ are parameter matrices; $d_m$ is the input dimension, while $d_k$ and $d_v$ are hidden dimensions of each projection subspace; h is the head number; $A \in \mathbb{R}^{l \times l}$ is the attention matrix and $l$ is the sequence length.

**FFN** consists of two cascaded linear transformations with a ReLU activation in between:

$$
\text{FFN}(x) = \max\left(0, xW_1 + b_1\right) W_2 + b_2,
\tag{2}
$$

where $W_1$ and $W_2$ are weights of two linear layers, while $b_1$ and $b_2$ are corresponding biases.

**LN** is applied before each layer of MSA and FFN, and the transformed $\hat{x}$ is calculated by:

$$
\hat{x} = x + [\text{MSA} \mid \text{FFN}](\text{LN}(x)).
\tag{3}
$$

## 4   EA-based Transformer

In this section, we firstly expand the association between operators in EA and modules in Transformer, and derive a unified mathematical representation for each set of corresponding pairs, expecting an evolutionary explanation for *why Transformer architecture works*. Then we introduce the SFC module to sequence 2D image that conforms to standard NLP format. Thus we may only focus on designing one unified model to solve multi-modal data. Finally, an EA-based Transformer only containing 1D operators is proposed for vision tasks, and we argue that this property is more suitable for hardware optimization in different scenarios.

### 4.1   EA Interpretation of Transformer Structure

Analogous to EA, the transformer structure has conceptually similar sub-modules, as shown in Figure 3. For both of methods, we define the individual (patch embedding) as $\boldsymbol{x}_i = [x_{i,1}, x_{i,2}, \ldots, x_{i,d}]$, where $d$ indicates sequence depth. Denoting $l$ as the sequence length, the population (patch embeddings) can be defined as $X = [\boldsymbol{x}_1, \boldsymbol{x}_2, \ldots, \boldsymbol{x}_l]^{\mathrm{T}}$.

**Crossover Operator** *vs*. **SA Module**. In EA, the crossover operator aims at creating new individuals by combining parts of other individuals. In detail, for an individual $\boldsymbol{x}_1 = [x_{1,1}, x_{1,2}, \ldots, x_{1,d}]$, the operator will randomly pick another individual $\boldsymbol{x}_i = [x_{i,1}, x_{i,2}, \ldots, x_{i,d}](1 \leq i \leq l)$ in the global population, and then randomly replaces features of $\boldsymbol{x}_1$ with $\boldsymbol{x}_i$ to form the new individual $\hat{\boldsymbol{x}}_1$:

$$
\hat{\boldsymbol{x}}_{1,j} = \begin{cases} \boldsymbol{x}_{i,j} & if\ randb(j) \leqslant CR \\ \boldsymbol{x}_{1,j} & otherwise \end{cases}, \quad s.t.\ j = 1, 2, \ldots, d,
\tag{4}
$$

where $randb(j)$ is the $j$-th evaluation of a uniform random number generator with outcome $\in [0, 1]$, $CR$ is the crossover constant $\in [0, 1]$ that is determined by the user. We re-formulate this process as:

$$
\begin{aligned}
\hat{\boldsymbol{x}}_1 &= \boldsymbol{w}_1 \cdot \boldsymbol{x}_1 + \boldsymbol{w}_i \cdot \boldsymbol{x}_i \\
&= \boldsymbol{w}_1 \cdot \boldsymbol{x}_1 + 0 \cdot \boldsymbol{x}_2 + \cdots + \boldsymbol{w}_i \cdot \boldsymbol{x}_i + \cdots + 0 \cdot \boldsymbol{x}_l \\
&= W_1^{cr} X_1 + 0X_2 + \cdots + W_i^{cr} X_i + \cdots + 0X_l,
\end{aligned}
\tag{5}
$$

where $\boldsymbol{w}_1$ and $\boldsymbol{w}_i$ are vectors filled with zeros or ones, representing the selection of different features of $\boldsymbol{x}_1$ and $\boldsymbol{x}_i$. $W_1^{cr}$ and $W_i^{cr}$ are corresponding diagonal matrix representations.

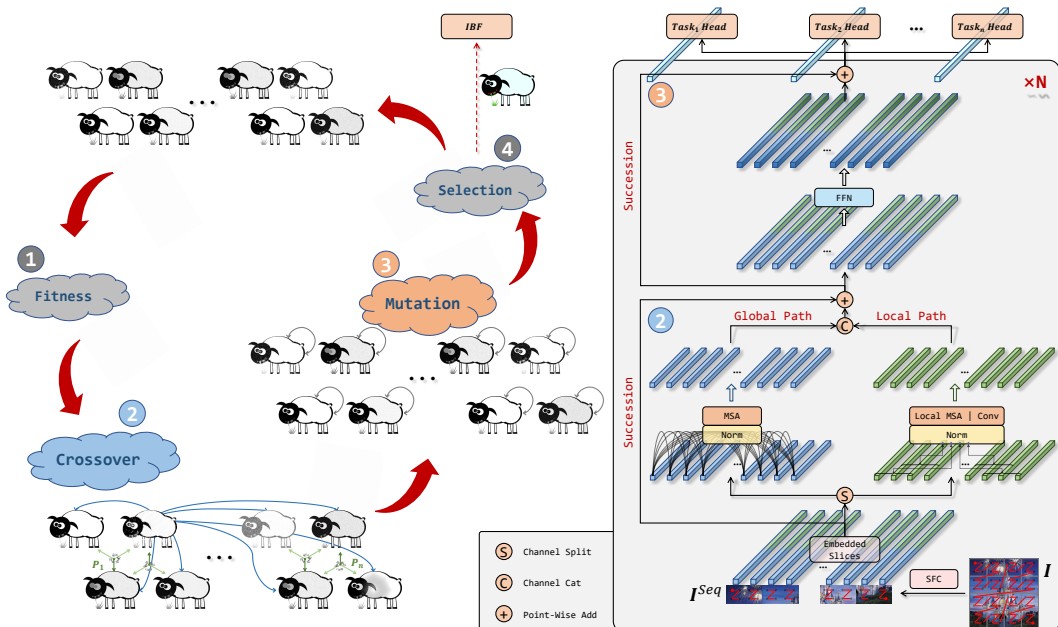

Figure 3: Structure of the proposed EAT. The right part illustrates the detailed procedure inspired by the left evolutionary algorithm. The blue and green lines in EA represent information interactions among individuals in global and local populations, which feeds back the design for global and local paths in our EAT. $P_1, \cdots, P_n$ represent biological evolution inside each local population. The proposed SFC module maps the 2D image into a 1D sequence so that only 1D operators are required.

For the SA module in transformer (MSA degenerates to SA when there is only one head), each patch embedding interacts with all embeddings. Without loss of generality, $\boldsymbol{x}_1 = [x_{1,1}, x_{1,2}, \ldots, x_{1,d}]$ interacts with the whole population, *i.e.*, $X$, as follows:

$$
\begin{aligned}
\hat{\boldsymbol{x}}_1 &= A_1 V_1 + A_2 V_2 + \cdots + A_l V_l \\
&= A_1 W^V X_1 + A_2 W^V X_2 + \cdots + A_l W^V X_l,
\end{aligned} \tag{6}
$$

where $A_i (i = 1, 2, \cdots, l)$ are attention weights of all patch embedding tokens that are calculated from queries and keys, $W^V$ is the parameter metric for the value projection. By comparing Equations 5 with 6, we find that they share the same formula representation, and the crossover operation is a sparse global interaction while SA has more complex computing and modeling capabilities.

**Mutation Operator** *vs*. **FFN Module**. In EA, the mutation operator injects randomness into the population by stochastically changing specific features of individuals. For an individual $\boldsymbol{x}_i = [x_{i,1}, x_{i,2}, \ldots, x_{i,d}](1 \leq i \leq l)$ in the global population, it goes through the mutation operation to form the new individual $\hat{\boldsymbol{x}}_i$, formulated as follows:

$$
\hat{x}_{i,j} = \begin{cases} rand(v_j^L, v_j^H) x_{i,j} & if\ randb(j) \leqslant MU \\ x_{i,j} & otherwise \end{cases}, \ s.t.\ j = 1, 2, \ldots, d, \tag{7}
$$

where the $MU$ is the mutation constant $\in [0, 1]$ that the user determines, $v_j^L$ and $v_j^H$ are lower and upper scale bounds of the $j$-th feature. Similarly, we re-formulate this process as:

$$
\begin{aligned}
\hat{\boldsymbol{x}}_i &= \boldsymbol{w}_i \cdot \boldsymbol{x}_i = [w_{i,1} x_{i,1}, w_{i,2} x_{i,2}, \cdots, w_{i,l} x_{i,l}] \\
&= W_i^{mu} X_i,
\end{aligned} \tag{8}
$$

where $\boldsymbol{w}_i$ is a randomly generated vector that represents weights on different characteristic depths, while $W_i^{mu}$ is the corresponding diagonal matrix representation.

For the FFN module in transformer, each patch embedding carries on directional feature transformation through cascaded linear layers (*c.f.*, Equation 2). Take one-layer linear as an example:

$$
\hat{\boldsymbol{x}}_i = W_1^{FFN} X_i, \tag{9}
$$

where $W_1^{FFN}$ is the weight of the first linear layer of the FFN module, and it is applied to each position separately and identically. By comparing Equations 8 and 9, we also find that they have the

same formula representation. FFN module is more expressive because it contains cascaded linear layers and non-linear ReLU activation layers in between, as depicted in Equation 2.

**Population Succession** *vs*. **Residual Connection**. In the evolution of the biological population, individuals at the previous stage have a certain probability of inheriting to the next stage. This phenomenon is expressed in the transformer in the form of residual connection, *i.e.*, patch embeddings of the previous layer are directly mapped to the next layer.

**Best Individual** *vs*. **Task-Related Token**. EAT chooses the enhanced task-related token (*e.g.*, classification token) associated with all patch embeddings as the target output, while EA chooses the individual with the best fitness score among the population as the object.

**Necessity of Modules in Transformer.** As described in the work [30], the absence of the Crossover operator or Mutation operator will significantly damage the model's performance. Similarly, Dong *et al*. [22] explore the effect of MLP in Transformer and find that MLP stops the output from degeneration. Furthermore, removing MSA in Transformer would also greatly damage the effectiveness of the model. Thus we can conclude that global information interaction and individual evolution are necessary for Transformer, just like the global Crossover and individual Mutation in EA.

### 4.2 Detailed Architecture of EAT

**Local Path.** Inspired by works [44, 14, 41, 63, 59] that introduce local and dynamic population concept to the evolutionary algorithm, we analogically introduce a local operator into the naive transformer structure in parallel with global MSA operation. As shown in Figure 3, the input features are divided into global features (marked blue) and local features (marked green) at the channel level with ratio $p$, which are then fed into global and local paths to conduct feature interaction, respectively. The outputs of the two paths recover the original data dimension by concatenation operation. Thus the improved module is very flexible and can be viewed as a plug-and-play module for the current transformer structure. This structure also implicitly introduces the design of multi-modal fusion, analogous to grouping convolution with the group equaling two. Specifically, the local operator can be 1D convolution, local MSA, 1D DCN [84], *etc*. In this paper, we adopt 1D convolution as the local operator that is more efficient and parameter friendly, and the improved transformer structure owns a constant number of sequentially executed operations and $O(1)$ maximum path length between any two positions, *c.f*., Appendix A. Therefore, the proposed structure maintains the same parallelism and efficiency as the original vision transformer structure. And $p$ is set to 0.5 for containing the minimal network parameters and FLOPs, proved by Proposition 1 and Proposition 2 in Appendix B.

**Proposition 1.** *The numbers of global and local branches in the $i-th$ mixed attention layer ($MA^i$) is $d_1$ and $d_2$. For the $i-th$ input sequence $F^i \in \mathbb{R}^{l \times d}$, the parameter of $MA^i$ is minimum when $d_1 = d_2 = d/2$, where $l$ and $d$ are sequence length and dimension, respectively.*

**Proposition 2.** *The numbers of global and local branches in the $i-th$ mixed attention layer ($MA^i$) is $d_1$ and $d_2$. For the $i-th$ input sequence $F^i \in \mathbb{R}^{l \times d}$, the FLOPs of $MA^i$ is minimum when $d_1 = d/2 + l/8, d_2 = d/2 - l/8$, where $l$ and $d$ are sequence length and dimension, respectively.*

**Task-Related Head.** Furthermore, we remove the Cls token that has been used since ViT [23], and propose a new paradigm to vision transformer that uses the task-related Head to obtain the corresponding task output through the final features. In this way, the transformer-based backbone can be separated from specific tasks. Each transformer layer does not need to interact with the Cls token, so the computation amount reduces slightly from $O((n+1)^2)$ to $O(n^2)$. Specifically, we use cross-attention to implement this head module. As shown in Figure 4, $K$ and $V$ are output features extracted by the transformer backbone, while $Q$ is the task-related token that integrates information through cross-attention. $M$ indicates that each Head contains M layers and is set to 2 in the paper. This design is flexible, with a negligible computation amount compared to the backbone.

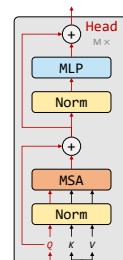

Figure 4: Structure of the proposed task-related Head.

**Convergence Analysis.** The convergence of Adam has been well studied in previous work [55]. It can be verified that global path and local path are both Lipschitz-smooth and gradient-bounded, as long as the transformer embedding is Lipschitz-smooth and gradient-bounded. In this case, the iteration sequence of the training algorithm converges to a stationary point of the learning objective with a convergence rate $O(1/sqrt(T))$, where $T$ is the number of iterations.

### 4.3 Space-Filling Curve Module

In this subsection, we mainly deal with two-dimensional SFCs. The mathematical definition of SFC is a surjective function that continuously maps a closed unit interval:

$$\omega = \{z \mid 0 \leq z \leq 1\} = [0, 1], \text{ where } z \in \omega \tag{10}$$

as the domain of the function and is mapped onto the unit square:

$$\Omega = \{(x, y) \mid 0 \leq x \leq 1, \quad 0 \leq y \leq 1\}, \text{ where } (x, y) \in \Omega \tag{11}$$

which is our range of the function. One can easily define a surjective mapping from $\omega$ onto $\Omega$, but it is not injective [67], *i.e.*, not bijective. Nevertheless, for a finite 1D sequence that maps 2D pixels, it contains bijective property.

**Remark 1.** *According to the definition of SFC, it must be surjective. For a finite 1D to 2D mapping, there is bound to be an intersection if the SFC is not injective, where there are at least two 1D points mapping to one same 2D point, i.e., $z_1 = SFC(x, y)$ and $z_2 = SFC(x, y)$. This is in contradiction to the finite SFC filling definition. Therefore, a finite SFC is bijective, and we can carry out two-way data lossless transformation through the pre-defined SFC.*

We propose an SFC module to map 2D images to the uniform 1D sequential format based on the above content. Therefore, we can address sequence inputs to handle multi-modal tasks in *one* unified model, *e.g.*, TIR and VLN. Taking image sequence as an example. As shown in the right bottom of Figure 3, the input image $\boldsymbol{I} \in \mathbb{R}^{H \times W \times 3}$ goes through the SFC module to be $\boldsymbol{I}^{Seq} \in \mathbb{R}^{HW \times 3}$, formulated as:

$$\boldsymbol{I}_k^{Seq} = \boldsymbol{I}_{i,j}, \text{ where } k = SFC(x, y), \tag{12}$$

where $1 \leq i \leq H$, $1 \leq j \leq W$, $1 \leq k \leq HW$, and $SFC$ can be any manually specified space-filling curve. The difference among SFCs mainly reflects how the 1D sequence preserves the 2D spatial structure. As an extension, SFC can be defined to three or higher dimensional data space and can also serialize the intermediate network features, *e.g.*, the output feature maps of stage 3 in ResNet [31]. In this paper, we use

Figure 5: Indexing illustration of the proposed SweepInSweep SFC.

*Embedded Slices* instead of *Embedded Patches* as the input of backbone that is consistent with NLP input format. As shown in Figure 5, we further propose a new SFC named SweepInSweep (SIS) that is equivalent to the input mode of the existing transformer-based methods, *e.g.*, ViT and DeiT, where the slice size equaling 16 is equivalent to that patch size equaling 4. The paper uses this mode for SFC by default. More visualizations of different SFCs are presented in Appendix C.

## 5 Experiments

### 5.1 Implementation Details.

We choose SOTA DeiT [64] as the baseline and follow the same experimental setting. By default, we train each model for 300 epoch from scratch without pre-training and distillation. The classification task is evaluated in ImageNet-1k dataset [20], and we conduct all experiments on a single node with 8 V100 GPUs. TIR is conducted on Fashion200k [27], MIT-States [34], and CSS [71] datasets, while VLN is performed on the R2R navigation dataset [1]. Detailed EAT variants can be viewed in Appendix D, and we supply the train and test codes in the supplementary material.

### 5.2 Comparison with SOTA Methods

The classification task is conducted on the ImageNet-1k dataset without external data. As noted in Table 1, we list four kinds of EAT of different magnitudes with *backbone + head* format, *i.e.*, Tiny, Small, Middle, and Base. They have comparable parameters with SOTA DeiT, but *a higher accuracy, a lower inference memory occupancy, and more throughput* in both GPU (tested with batch size equaling 256 in single V100) and CPU (tested with batch size equaling 16 in i7-8700K @3.70GHz). Our method is trained in 224 resolution for 300 epochs without distillation, so our result is slightly lower than large CNN-based EfficientNet in terms of Top-1. Moreover, we visualize comprehensive Top-1 results of different methods under various evaluation indexes in Figure 6 and present the results of a series of EAT models with different model sizes. *1)* Left two subgraphs show that our EAT outperforms the strong baseline DeiT, and even obtains better results than EfficientNet in both GPU and CPU throughput, *e.g.*, EAT-M. *2)* Besides, our approach is very competitive for

Table 1: Comparison with SOTA CNN-based and Transformer-based methods on ImageNet-1k dataset. Reported results are from corresponding papers. 🐦: Distillation; 1ke: Training 1000 epochs.

| Network | Params. ↓ (M) | GFlops. ↓ | Images/s ↑ GPU | Images/s ↑ CPU | Image Size | Top-1 | Inference Memory |
|---|---|---|---|---|---|---|---|
| *CNN-Based Nets* | | | | | | | |
| ResNet-18 | 11.7M | 1.8 | 4729 | 77.9 | $224^2$ | 69.8 | 1728 |
| ResNet-50 | 25.6M | 4.1 | 1041 | 21.1 | $224^2$ | 76.2 | 2424 |
| ResNet-101 | 44.5M | 7.8 | 620 | 13.2 | $224^2$ | 77.4 | 2574 |
| ResNet-152 | 60.2M | 11.5 | 431 | 9.4 | $224^2$ | 78.3 | 2694 |
| RegNetY-4GF | 20.6M | 4.0 | 976 | 22.1 | $224^2$ | 80.0 | 2828 |
| RegNetY-8GF | 39.2M | 8.0 | 532 | 12.9 | $224^2$ | 81.7 | 3134 |
| RegNetY-16GF | 83.6M | 15.9 | 316 | 8.4 | $224^2$ | 82.9 | 4240 |
| EfficientNet-B0 | 5.3M | 0.4 | 2456 | 49.4 | $224^2$ | 77.1 | 2158 |
| EfficientNet-B2 | 9.1M | 1.0 | 1074 | 22.6 | $260^2$ | 80.1 | 2522 |
| EfficientNet-B4 | 19.3M | 4.5 | 313 | 6.1 | $380^2$ | 82.9 | 5280 |
| EfficientNet-B5 | 30.4M | 10.4 | 145 | 3.0 | $456^2$ | 83.6 | 7082 |
| EfficientNet-B7 | 66.3M | 38.2 | 48 | 1.0 | $600^2$ | 84.3 | 14650 |
| NFNet-F0 | 71.5M | 9.6 | 574 | 12.5 | $256^2$ | 83.6 | 2967 |
| *Transformer-Based Nets* | | | | | | | |
| ViT-B/16 | 86.6M | 17.6 | 291 | 10.3 | $384^2$ | 77.9 | 2760 |
| ViT-L/16 | 304M | 61.6 | 92 | 2.7 | $384^2$ | 76.5 | 4618 |
| DeiT-Ti | 5.7M | 1.3 | 2437 | 89.8 | $224^2$ | 72.2 | 1478 |
| DeiT-S | 22.1M | 4.6 | 927 | 31.4 | $224^2$ | 79.8 | 1804 |
| DeiT-B | 86.6M | 17.6 | 290 | 10.2 | $224^2$ | 81.8 | 2760 |
| EAT-Ti | 5.7M ( 4.8M + 0.9M) | 1.0 | 2442 | 95.4 | $224^2$ | 72.7 | 1448 |
| EAT-S | 22.1M (18.5M + 3.6M) | 3.8 | 1001 | 34.4 | $224^2$ | 80.4 | 1708 |
| EAT-M | 49.0M (41.0M + 8.0M) | 8.4 | 519 | 18.4 | $224^2$ | 82.1 | 2114 |
| EAT-B | 86.6M (72.4M +14.2M) | 14.8 | 329 | 11.7 | $224^2$ | 82.0 | 2508 |
| DeiT-Ti 🐦 | 5.9M | 1.3 | 2406 | 87.1 | $224^2$ | 74.5 | 1476 |
| DeiT-Ti / 1ke 🐦 | 5.9M | 1.3 | 2406 | 87.1 | $224^2$ | 76.6 | 1476 |
| EAT-Ti 🐦 | 5.7M ( 4.8M + 0.9M) | 1.0 | 2442 | 95.4 | $224^2$ | 74.8 | 1448 |
| EAT-Ti / 1ke 🐦 | 5.7M ( 4.8M + 0.9M) | 1.0 | 2442 | 95.4 | $224^2$ | 77.0 | 1448 |

resource-restrained scenarios, *e.g.*, having a lower inference memory occupancy and parameters while obtaining considerable accuracy. *3)* Also, we find accuracy saturation in transformer structure that is also mentioned in recent works [65, 82], and we look forward to follow-up works to solve this problem. Furthermore, some techniques to help improve the accuracy of the model are applied, and our EAT model obtains a better accuracy.

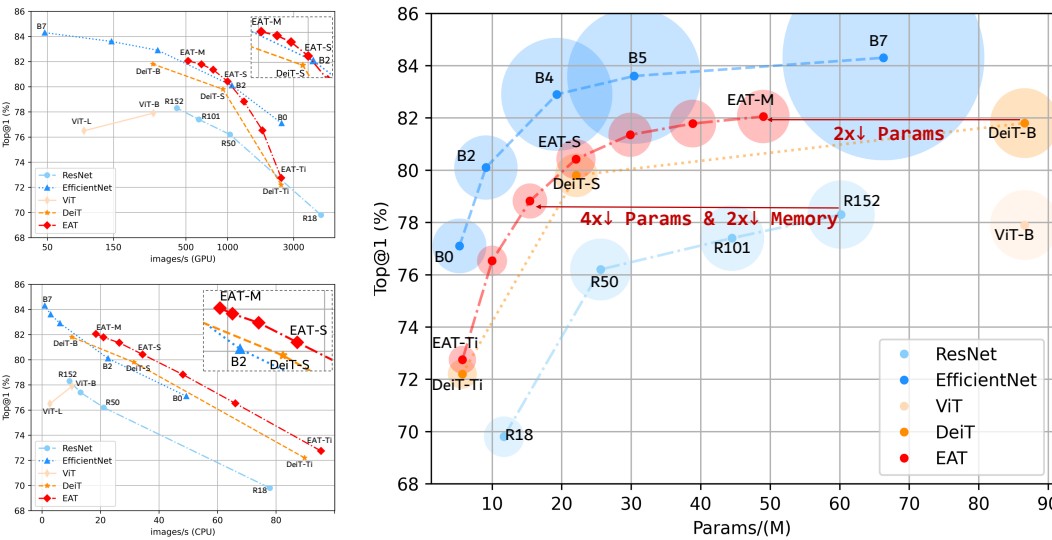

Figure 6: Comparison of different methods under different evaluation indexes. From left to right are GPU throughput, CPU throughput, and model parameters. The smaller the circle in the right sub-figure, the smaller the inference memory occupancy. Please zoom in for a better visual effect.

Table 2: Retrieval performance on three datasets in rank-1. The vision and language models are replaced by other nets, marked in the lower right footnote as $V$ and $L$. $\star$: Our reproduction; -: No corresponding result. The best number is in bold.

| Method | CSS | MIT-States | Fashion200k |
|---|---|---|---|
| TIRG [71] | - | 12.2 | 14.1 |
| TIRG$^\star$ | 70.1 | 13.1 | 14.0 |
| + TR$_L$ | 70.7 | 13.2 | 14.3 |
| + EAT$_L$ | 71.0 | 13.4 | 14.9 |
| + R101$_V$ | 73.0 | 14.3 | 19.0 |
| + EAT$_V$ | 73.5 | 14.9 | 19.9 |
| + EAT$_{V+L}$ | **73.8** | **15.0** | **20.1** |

Table 3: VLN experiment on the R2R dataset. $\star$: Our reproduction with suggested parameters by authors. NE: Navigation Error; SPL: Success weighted by (normalized inverse) Path Length; nDTW: normalized Dynamic Time Warping.

| | Method | NE $\downarrow$ | SPL $\uparrow$ | nDTW $\uparrow$ |
|---|---|---|---|---|
| Seen | PTA$^\star$ | 3.98 | 0.61 | 0.71 |
| | + EAT$_L$ | **3.84** | **0.62** | **0.72** |
| | + EAT$_V$ | 3.95 | 0.61 | 0.71 |
| | + EAT$_{V+L}$ | 3.95 | 0.61 | 0.71 |
| Unseen | PTA$^\star$ | 6.61 | 0.28 | 0.48 |
| | + EAT$_L$ | **6.41** | **0.33** | **0.51** |
| | + EAT$_V$ | 6.63 | 0.29 | 0.49 |
| | + EAT$_{V+L}$ | 6.43 | 0.32 | **0.51** |

Table 4: Ablation study for several items on ImageNet-1k. *Default* represents the baseline method based on EAT-B. The gray font indicates that the corresponding parameter is not be modified.

| Ablation Items | Head Layers | Local Ratio | FFN Ratio | Top-1 | Ablation Items | Kernel Size | Local Operator | SFC Mode | Image Size | Top-1 |
|---|---|---|---|---|---|---|---|---|---|---|
| Default | 1 | 0.50 | 4 | 80.264 | | 3 | 1D Conv | SIS | $224^2$ | 80.264 |
| Head Layer | 0 | 0.50 | 4 | 79.692 | Kernel Size | 1 | 1D Conv | SIS | $224^2$ | 79.128 |
| | 2 | 0.50 | 4 | 80.422 | | 5 | 1D Conv | SIS | $224^2$ | 80.256 |
| | 3 | 0.50 | 4 | 80.435 | | 7 | 1D Conv | SIS | $224^2$ | 80.052 |
| | 4 | 0.50 | 4 | 80.454 | Local Operator | 3 | Local MSA | SIS | $224^2$ | 79.940 |
| | 5 | 0.50 | 4 | 80.446 | | 3 | DCN | SIS | $224^2$ | 68.070 |
| Local Ratio | 1 | 0.25 | 4 | 80.280 | SFC Mode | 3 | 1D Conv | Sweep | $256^2$ | 71.280 |
| | 1 | 0.75 | 4 | 79.518 | | 3 | 1D Conv | Scan | $256^2$ | 74.164 |
| FFN Ratio | 1 | 0.50 | 2 | 77.422 | | 3 | 1D Conv | Hilbert | $256^2$ | 79.842 |
| | 1 | 0.50 | 3 | 79.176 | | 3 | 1D Conv | Z-Order / SIS | $256^2$ | 80.438 |

## 5.3 Multi-Modal Experiments

**Text-based Image Retrieval.** TIR is the task of searching for semantically matched images in a large image gallery according to the given search query, *i.e.*, an image and a text string. Table 2 illustrates the quantitative results. Naive transformer and EAT$_L$ have the same number of layers, and EAT$_V$ has similar parameters as R101$_V$. The results show that EAT brings positive benefits by replacing either original sub-network, even though vision and language models share one structure.
**Vision Language Navigation.** We further assess our approach by VLN experiments, where an agent needs to follow a language-specified path to reach a target destination with the step-by-step visual observation. We choose the PTA [39] as the base method, and Table 3 shows results under the same experimental setting. Our unified EAT obtains comparable results with PTA in the seen mode but achieves greater improvement in the unseen mode, meaning better robustness. When only replacing the vision EAT, the result changes very little. We analyze the reason that the used visual feature dimension of EAT-B is smaller than the original ResNet-152 (from 2048 to 384), and reinforcement learning is sensitive to this. Overall, our unified EAT still contributes to the PTA method.

## 5.4 Ablation Study

Table 4 shows ablation results for several items on ImageNet-1k. *1) Head Layer.* Within a certain range, the model performance increases with the increase of the head layer, and it equals 2 in the paper for trading off model effectiveness and parameters. *2) Local Ratio.* Large local ratio causes performance falling and increases the parameters (*c.f.*, Section 4.2), so $p$ equaling 0.5 is a better choice. *3) FFN Ratio.* Similar to the conclusion of work [69], a lower FFN ratio results in a drop in accuracy. *4) Kernel Size.* Large kernel can lead to accuracy saturation, so we set it to 3 in the paper. *5) Local Operator.* We replace the local operator with local MSA and 1D DCN. The former slightly reduces the accuracy (a structure similar to the global path may result in learning redundancy), while the latter greatly reduces the accuracy (possibly hyper-parameters are not adapted). *6) SFC Mode.* Since some SFCs, *e.g.*, Hilbert and Z-Order, only deal with images with the power of side lengths, we set the image size to 256 here. Results show that the images serialized by Z-Order or SIS get better results because they can better preserve 2D spatial information that is important for 1D-convolution.

### 5.5 Model Interpretation by Attention Visualization

We average attention maps of each head layer and visualize them to illustrate which parts of the image the model is focusing on. As shown in Figure 7, Head1 pays more attention to subjects that are meaningful to the classification results, while the deeper Head2 integrates features of Head1 to form the final vector for classification that focuses on more comprehensive areas. Also, Grad-CAM [58] is applied to highlight concerning regions by our model and results consistently demonstrate the effectiveness of our proposed Head module. Note that the Cls Head contains two head layers, denoted as Head1 and Head2 in the figure. We also visualize attention maps of middle layers in Appendix F and find that the global path prefers to model long-distance over DeiT.

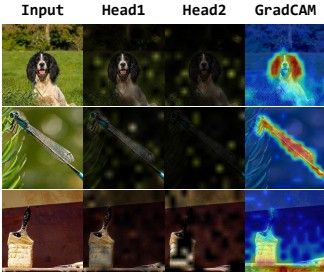

Figure 7: Visualization of attention maps for head layers and Grad-CAM results.

## 6 Conclusions

This paper explains the rationality of vision transformer by analogy with EA, which brings inspiration for the neural architecture design. Besides, the introduced SFC module serializes the multi-modal data into a sequential format, and we achieve the paradigm of using one unified model to address multi-modal tasks. Abundant experimental results demonstrate the effectiveness and flexibility of our approach. We hope this work may bring some enlightenment to network interpretability and thinking to design the unified model for multi-modal tasks. Recently, we observe that EA has been successfully used in other areas, *e.g.*, RL (PES [57], ERL [37]), NAS (SPOS [25], CARS [78]), Hyperparameter Adjustment (PBT [35]), *etc*. Nevertheless, how to combine heuristic algorithms such as EA with other fields is still a difficult problem, and we will further explore potential possibilities in the future.

**Acknowledgment** We thank all authors for their excellent contributions as well as anonymous reviewers and chairs for their constructive comments. This work is partially supported by the National Natural Science Foundation of China (NSFC) under Grant No. 61836015.

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
