# A  Properties of Different Layers

Table 5: Parameters (Params), floating point operations (FLOPs), Sequential Operations, and Maximum Path Length for different layer types. Assume that the input data is a sequence of length $l$ and depth $d$. $k$ is the kernel size for 1D convolution and local self-attention, $n$ is the query length, and $inf$ stands for infinity.

| Layer Type | Params | FLOPs | Sequential Operations | Maximum Path Length |
|---|---|---|---|---|
| Linear | $(d+1)d$ | $2d^2l$ | $O(1)$ | $O(inf)$ |
| 1D Convolution | $(kd+1)d$ | $2d^2lk$ | $O(1)$ | $O(n/k)$ |
| Self-Attention | $4(d+1)d$ | $8d^2l + 4dl^2 + 3l^2$ | $O(1)$ | $O(1)$ |
| Local Self-Attention | $4(d+1)d$ | $8d^2l + 4dlk + 3ld$ | $O(1)$ | $O(n/k)$ |
| Cross-Attention | $4(d+1)d$ | $4d^2l + (4dl + 2d^2 + 3l)n$ | $O(1)$ | $O(1)$ |

We adopt 1D convolution as the local operator that is more efficient and parameter friendly, and the improved transformer structure owns a constant number of sequentially executed operations and $O(1)$ maximum path length between any two positions.

# B  Analysis of the Local Ratio

**Proposition 3.** *The numbers of global and local branches in the $i-th$ mixed attention layer ($MA^i$) is $d_1$ and $d_2$. For the $i-th$ input sequence $F^i \in \mathbb{R}^{l\times d}$, the parameter of $MA^i$ is minimum when $d_1 = d_2 = d/2$, where $l$ and $d$ are sequence length and dimension, respectively.*

*Proof.* Given the $F^i \in \mathbb{R}^{l\times d}$ and $MA^i$ with $d_1$ and $d_2$, the overall parameters $Params^i = 4(d_1 + 1)d_1 + (d_2 + 1)d_2 + (kd_2 + 1)d_2$ according to Table 5 ($k$ is the kernel size), and it is factorized as follows:

$$Params^i = 4(d_1 + 1)d_1 + (d_2 + 1)d_2 + (kd_2 + 1)d_2$$
$$= 4d_1{}^2 + (k+1)d_2{}^2 + 4d_1 + 2d_2 \tag{13}$$

Based on $d_1 + d_2 = d$, we have

$$Params^i = (5+k)d_2{}^2 - (8d+2)d_2 + 4d^2 + 4d \tag{14}$$

Applying the minimum value formula of a quadratic function to the equation 14, we can obtain the minimum value $2d^2 + 3d + 1/8$, where $d_1 = d/2 - 1/8$ and $d_2 = d/2 + 1/8$. Given that $d_1$ and $d_2$ are integers, we make $d_1 = d_2 = d/2$. Therefore, the minimum value of equation 14 becomes $2d^2 + 3d$ that is nearly half of the original self-attention layer, *i.e.*, $4d^2 + 4d$, according to Table 5. □

**Proposition 4.** *The numbers of global and local branches in the $i-th$ mixed attention layer ($MA^i$) is $d_1$ and $d_2$. For the $i-th$ input sequence $F^i \in \mathbb{R}^{l\times d}$, the FLOPs of $MA^i$ is minimum when $d_1 = d/2 + l/8, d_2 = d/2 - l/8$, where $l$ and $d$ are sequence length and dimension, respectively.*

*Proof.* Given the $F^i \in \mathbb{R}^{l\times d}$ and $MA^i$ with $d_1$ and $d_2$, the overall FLOPs $FLOPs^i = 8d_1{}^2l + 4d_1l^2 + 3l^2 + 2d_2{}^2l + 2d_2{}^2lk$ according to Table 5 ($k$ is the kernel size), and it is factorized as follows:

$$FLOPs^i = 8d_1{}^2l + 4d_1l^2 + 3l^2 + 2d_2{}^2l + 2d_2{}^2lk$$
$$= (8d_1{}^2 + 4d_1l + 3l + 2d_2{}^2 + 2d_2{}^2k)l \tag{15}$$

Based on $d_1 + d_2 = d$, we have

$$FLOPs^i = [(10 + 2k)d_2{}^2 - (16d + 4l)d_2 + 8d^2 + 4dl + 3l]l \tag{16}$$

Applying the minimum value formula of a quadratic function to the equation 16, we can obtain the minimum value $(4d^2 + 2ld - l^2/4 + 3l)l$, where $d_1 = d/2 - l/8$ and $d_2 = d/2 + l/8$. Given that $d_1$ and $d_2$ are integers and $l$ is usually much smaller than $d$ in practice, we make $d_1 = d_2 = d/2$ that is consistent with the above proposition 1. Therefore, the minimum value of equation 16 becomes $(4d^2 + 2ld + 3l)l$ that is nearly half of the original self-attention layer according to Table 5, *i.e.*, $(8d^2 + 4ld + 3l)l$. □

## C   Visualization of Hilbert SFC

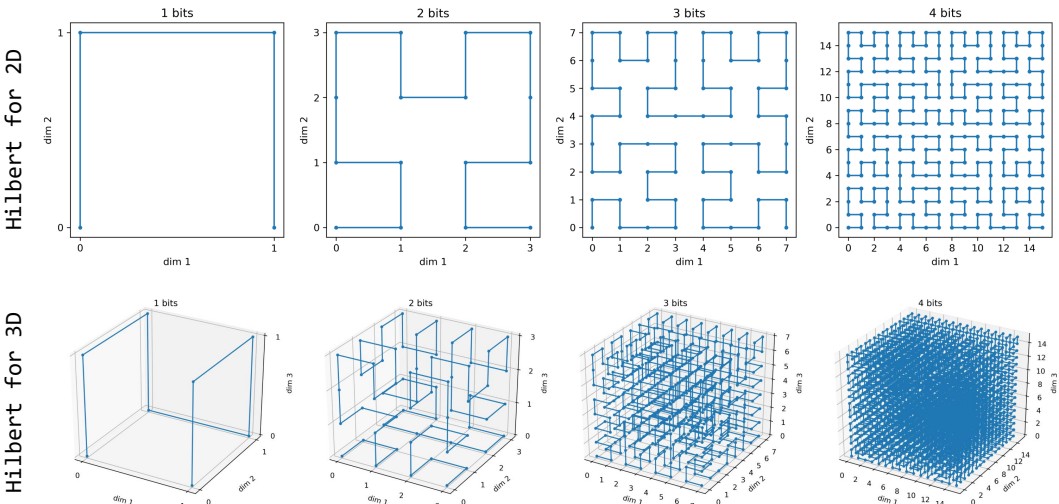

Figure 8: Visualization of Hilbert SFC in 2D and 3D format under different image bits.

We visualize the SFC of Hilbert for a more intuitive understanding in Figure 8.

## D   Variants of EAT

Table 6: Detailed settings of our EAT variants.

| Model | Emb. Dim. | H. in MSA | Layers | Head Layers | FFN Ratio | Local Ope. | Local Ratio | Kernel Size | SFC Mode | Image Size | Params. |
|---|---|---|---|---|---|---|---|---|---|---|---|
| EAT-Ti | 192 | 2 | 12 | 2 | 4 | 1D Conv | 3 | 0.5 | SIS | $224^2$ | 5.7M |
| EAT-S | 384 | 3 | 12 | 2 | 4 | 1D Conv | 3 | 0.5 | SIS | $224^2$ | 22.1M |
| EAT-M | 576 | 4 | 12 | 2 | 4 | 1D Conv | 3 | 0.5 | SIS | $224^2$ | 49.0M |
| EAT-B | 768 | 6 | 12 | 2 | 4 | 1D Conv | 3 | 0.5 | SIS | $224^2$ | 86.6M |

Table 6 shows detailed settings of our proposed four EAT variants.

## E   Visualization of Attention Map in Task-Related Head

Taking the classification Head as an example, we visualize the attention map in the Head to intuitively explain why the model works. Specifically, we choose EAT-S here, which contains two layers for the classification Head, and each Head contains eight heads in the inner cross-attention layer. Here, capital Head indicates task-related Head, while lowercase head represents multi-head in the cross-attention. As shown in Figure 9, we normalize values of attention maps to [0, 1] and draw them on the right side of the image. Results show that different heads focus on different regions in the image, and the deeper Head2 integrates features of Head1 to form the final vector for classification that focuses on a broader area.

Furthermore, we average eight attention maps of each head layer and use it as the global attention map that represents which parts of the image the corresponding head layer is focusing on, as shown in Figure 11. Also, Grad-CAM [58] is applied to produce a coarse localization map highlighting the crucial regions in the image. By analyzing results, we find that both visualization methods focus more on the subjects in the image, demonstrating the effectiveness of our proposed Head module.

## F   Visualization of Attention Map in Middle Layers

We also visualize some attention maps for middle layers, taking the heads in the fourth and sixth layers as an example. As shown in Figure 10, compared with DeiT without local modeling, our EAT pays more attention to global information fusion, where more significant values are found at off-diagonal locations. We analyze the reason for this phenomenon that the parallel local path takes responsibility for some of the local modelings that would have been the responsibility of the MSA.

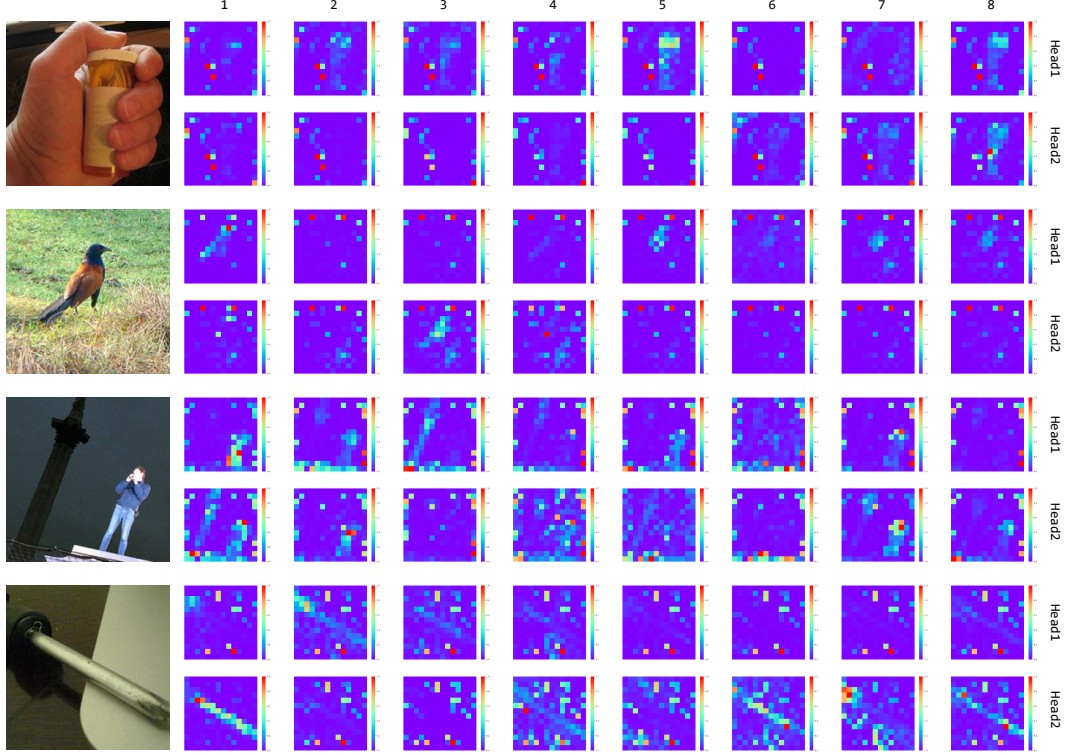

Figure 9: Visualization of Attention Maps in the classification head. We display two head layers for each image and eight attention maps for eight heads in each head layer.

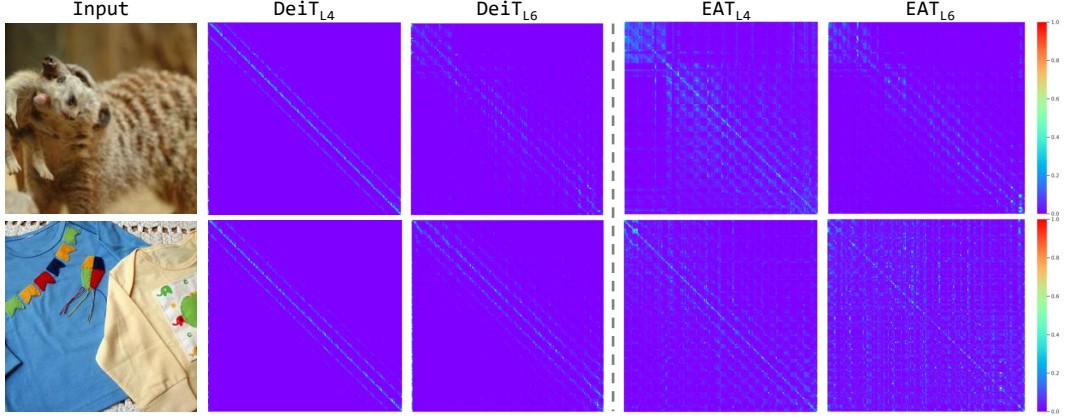

Figure 10: Visualization of attention maps for the fourth and sixth middle layers. The first column shows the input images; the second and third columns are visualized attention maps for DeiT, while the fourth and fifth columns for our EAT-S.

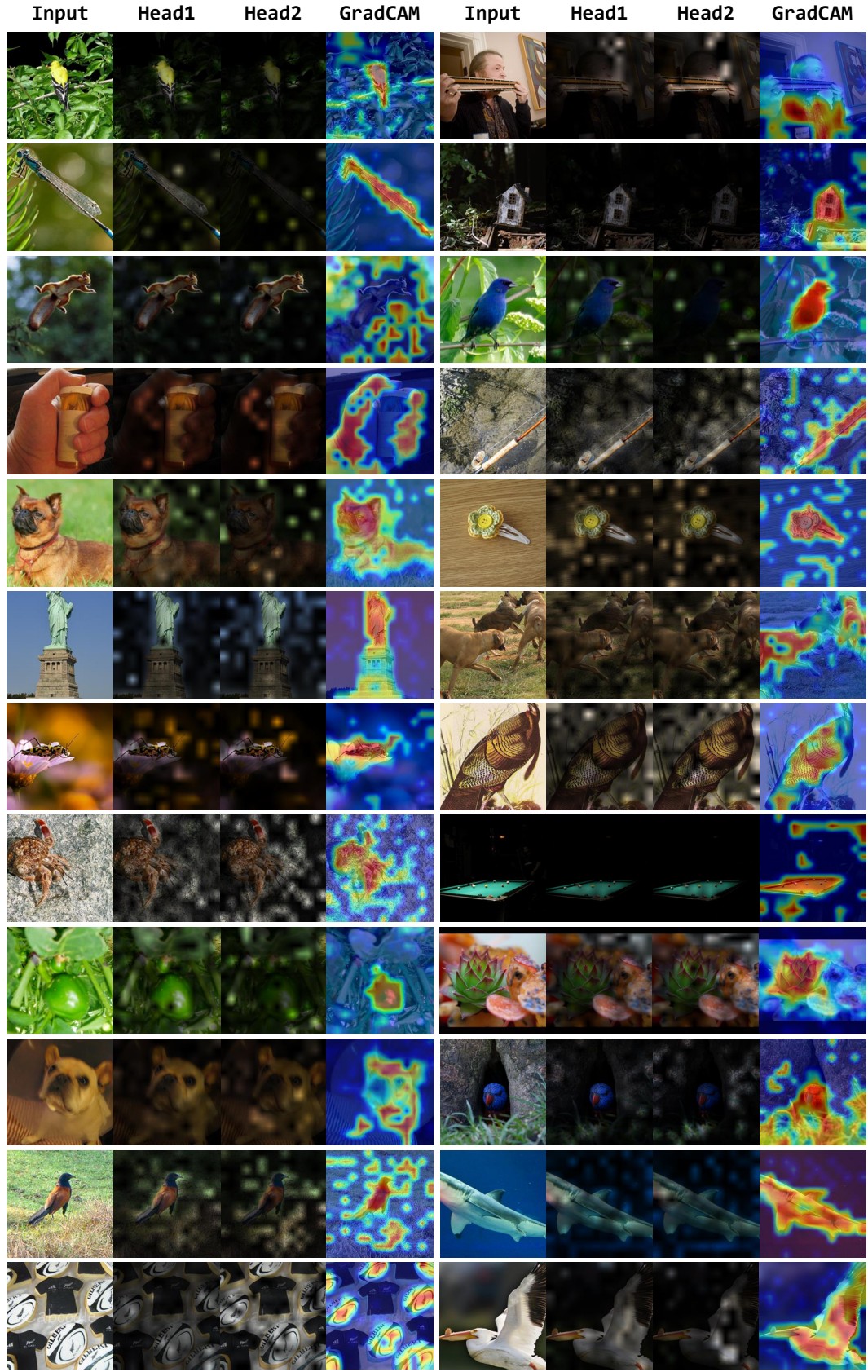

Figure 11: Visualization of overall attention for different head layers and Grad-CAM results.