# OpenReview forum: "Analogous to Evolutionary Algorithm: Designing a Unified Sequence Model"
_NeurIPS.cc/2021/Conference — NeurIPS 2021 Poster_

### Official Review · Reviewer_EQ9b · 2021-07-12

**Rating:** 7
**Confidence:** 5

**Summary:**

This paper presents a novel perspective to explains the rationality of vision transformer by analogy with a heuristic evolutionary algorithm and proposes an improved EAT model borrowing the dynamic local population concept from EA. Authors introduce the spatial-filling curve to serialize the input image, so only one unified model is required to handle multi-modal tasks. The extensive classification and multi-modal experiments show that the proposed method is effective and efficient.

**Limitations And Societal Impact:**

- Task-related head looks the same as the cross-attention structure in Transformer Decoder[61], and a similar structure is mentioned in Cait[57], so this contribution is optional.
- Spatial-filling curve does not seem to improve the performance, and the SIS mode stated in the paper is equivalent to the current token embedding in ViT.
- EAT increases by 0.5 points on average compared to the baseline in Table 1, but the reported highest Top-1 is 82.4 that is lower than current methods, e.g., SwinTransformer[36] and Cait[57]. Authors should compare with these methods or explain why it is not being compared.
- Why do authors claim that "... so that only 1D operators are required" in Figure 3? Adding 2D operations, such as 2D-convolution in CPVT[16], CeiT[70], and PVTv2[a], has been shown to improve the performance of Transformer in image classification tasks.
- Grammatical errors and typo issues lower the paper quality that can be seen in the following section.

[a] Wang W, Xie E, Li X, et al. PVTv2:

Moreover, Here are more suggestions.

- Some quantitative results in Appendix E should be presented in Table 1
- Why does DCN dramatically damage performance in Table 4?
- Figure 6 should be rearranged
- How to integrate CV and NLP features is unclear in multi-modal experiments, and more details should be displayed.
- Some grammar/typo issues.
    - L12, L51: multi-modal
    - L102: their ways
    - L116: the definition of "h"
    - L161: the weight
    - L174: concepts
    - L314: remove the comma after "format"




**Main Review:**

Strong Points
- The perspective between Transformer and EA is interesting, and the improved EAT is reasonable and effective. The introduced spatial-filling curve makes it possible to deal with multi-modal tasks by one unified model.
- The paper is easy to follow, and its organization is sound. The authors provide the training code, and the reported results are convincing.
- The experiments for classification and multi-modal tasks are extensive, and detailed ablation experiments are carried out.

**Time Spent Reviewing:**

5

---

> ### Author Response · Authors · 2021-08-10
> **Rebuttal by Authors for Reviewer EQ9b**
>
> Thanks to the reviewer for the comments and appreciation. We would like to clarify several things to address the reviewer's concerns:
>
> Q1: About Task-related Head
>
> A1: As stated in L200, we use cross-attention to implement this Task-related Head module, and it separates the transformer-based backbone from specific tasks more elegantly (*c.f.*, Sec. 4.2). Also, our module is different from CaiT [57] (using self-attention to realize the CA module) in that it is more efficient and has $7\times \downarrow$ smaller FLOPs than CA in CaiT (with dim=192).
>
> | **Method** | **Structure** | **FLOPs (M)** |
> |:-:|:-:|:-:|
> | Class-Attention (CaiT) | Self-Attention | 204.6|
> | Task-related Head (Ours) | Cross-Attention | **29.8** ($7\times \downarrow$)|
>
> Q2: About SFC
>
> A2: Space-Filling Curve (SFC) is designed to unify multi-modal input rather than to improve performance. SFC serves as a bridge between rasterized 2D image data and serialized 1D sequence data, making it possible to integrate the unified paradigm that uses a unified model to solve multi-modal data, e.g., the 2D image in CV and 1D sequence in NLP. Also, we propose a new SFC named SweepInSweep (SIS) that is equivalent to the current patch embeddings, successfully summarizing the current patch embeddings method into a unified SFC concept. Please refer to Reviewer Ppg5-A2 for a more detailed explanation.
>
> Q3: About the highest Top-1
>
> A3:
> - ***We focus on designing a unified and elegant paradigm to handle multi-modal tasks, rather than simply achieving higher scores on the classification task in this paper.*** Therefore, we use EA to explain TR in theory, introduce local path and task-related head to make the model more efficient and elegant, and implement a unified model to handle multi-modal tasks through SFC on the framework.
> - ***Our EAT is more efficient than the baseline DeiT*** and is designed with the same amount of parameters as the DeiT for relative comparison. Results in Table 1 show that EAT has faster speeds than DeiT (329 vs. 290, 13.4% $\uparrow$ for GPU and 11.7 vs. 10.2, 14.7% $\uparrow$ for CPU) and a smaller inference memory occupancy in Table 1. You can find more results under more robust settings and source code in the supplementary material.
> - EAT and DeiT belongs to columnar design in vision transformer, which differ structurally from the elaborate pyramid structure, so they are not strictly suitable for comparison.
> - Last but not least, our computing resource is limited, and we did not conduct experiments with bigger models, larger resolution, and longer time. For example, it takes more than a month to train DeiT-B (384 resolution; 1k epochs; distillation) with a single node with 8 V100 GPUs. Nevertheless, the results can be roughly deduced from the results of the DeiT and this paper.
>
>
> Q4: About 1D/2D operators
>
> A4: We will explain the necessity of the 1D operator for our unified paradigm to solve multi-modal tasks and its advantages.
> - In this paper, we aim at designing a unified model to solve multi-modal tasks. Inevitably, we need to unify multi-modal data as sequence data, so we only choose 1D operators to build our model.
> - Using only 1D operators makes the entire model more elegant, without the headaches of reshaping and permuting operations. Meanwhile, requiring only 1D operators means that the optimization workload of our method on different platforms will be very friendly, and only several 1d operators need to be optimized.
> - 2D operators indeed bring an essential role in many models, but it is not the object of our research and comparison in this paper.
>
> Q5: Some grammar/typo issues
>
> A5: We will seriously revise our final version.
>
> Q6: About quantitative results in Appendix E
>
> A6: Thanks for the suggestion, and we will improve this display.
>
> Q7: About DCN as the local operator
>
> A7: DCN did not train well, possibly because hyper-parameters are not adapted. We will open our source code and throw this question.
>
> Q8: Figure 6 should be rearranged
>
> A8: Thanks for the suggestion. We will rearrange the spatial position of Figure 6 in the final version.
>
> Q9: How to integrate CV and NLP features
>
> A9: The **git links** for multi-modal experiments can be viewed in the **README.md** in the supplementary source code. Specifically, image and text features are concatenated followed by several linear layers in the TIR, while image, text, and action information are concatenated followed by an RNN cell. We will fill in the details in the revised version.

---

### Official Review · Reviewer_Ppg5 · 2021-07-14

**Rating:** 6
**Confidence:** 4

**Summary:**

This paper attempts to explain the rationality of Vision Transformer by the proven practical Evolutionary Algorithm and finds that both of them have consistent mathematical representation. Besides, for separating the architecture of transformer-based network from the data format adaptation, the authors introduce the spatial-filling curve into the vision transformer to sequence multiple dimensional data (e.g., image or video) into a uniform sequential format. Thereby the proposed transformer-based network can deal with multi-modal tasks.

**Limitations And Societal Impact:**

The paper is easy to understand for me, but the writing should be checked carefully, such as “Imeges” in Table 1.

**Main Review:**

It is interesting that the theory of EA is introduced to explain the rationality of Vision Transformer. The authors give the comparison of the basic operator in EA with the module in transformer. It is concluded that the mathematical representations of Mutation operator with FFN Module, Crossover Operator with SA Module are consistent. Based on those observations, it is better to give readers more insightful explanations on the success of transformer instead of only stating those phenomenon.

Serializing the multiple dimensional data into sequence with a consistent algorithm is indeed an elegant solution to adapt multi-modal data for transformer. Many algorithms like those in Figure 2 can competent this task, and Hilbert algorithm is selected for serializing the input data. So, what is the reason behind for the authors to make this choice. It would be more convincing if those space filling curve algorithms are compared in the experiment section.


**Time Spent Reviewing:**

15.0 hours

---

> ### Author Response · Authors · 2021-08-10
> **Rebuttal by Authors for Reviewer Ppg5**
>
> Thanks to the reviewer for the comments and appreciation. We would like to clarify several things to address the reviewer's concerns:
>
> Q1: More insightful explanations
>
> A1: Thanks for the question. Besides the analogical explanations between EA and Transformer, we will further explain our approach in three aspects: 1) necessity of module; 2) visual interpretability; and 3) application in other areas.
> - As described in work [s16], the absence of the Crossover operator or Mutation operator will significantly damage the model's performance. Similarly, Dong~*et al*. [s17] explore the effect of MLP on TR and find that MLP stops the output from degeneration. Furthermore, removing ATT would also greatly damage the effectiveness of the model. Thus we can conclude that global information interaction and individual evolution are necessary for Transformer, just like the global Crossover and individual Mutation in EA.
> - Analogy with the dynamic local population, we design the local path in Transformer for modeling local feature interaction. We visualize the learned attention maps for DeiT and EAT in Appendix G. Results indicate that the ATT in our EAT prefers to model the long-distance relationship than DeiT, meaning that the global and local paths play different roles in modeling features at different distances. This proves the rationality and effectiveness of the introduced local path.
> - Recently, EA has been used successfully in other areas as well, e.g, RL (PES [50], ERL[s12]), NAS (SPOS [s13], CARS [s14]), Hyperparameter Adjustment (PBT [s15]). Nevertheless, how to combine heuristic algorithms such as EA with other fields is still a difficult problem, and we will continue to explore it. We believe there will be more and better works in the future.
>
> We will follow your advice and revise the way of writing in Introduction and Sec. 4.1 in the final version, aiming at giving more insightful explanations from EA to readers.
>
> Q2: About SFC
>
> A2: We will talk more about SFC in the following three points:
> - Figure 2 shows some instances of four SFCs, mainly for the illustrative purpose, since the concept of SFC is hardly mentioned or never used in deep learning.
> - SFC serves as a bridge between rasterized 2D image data and serialized 1D sequence data. For a fair comparison, we propose a new SIS SFC (Figure 5) that is equivalent to the patch embedding of the current Transformer-based method, and the default SFC mode is set to SIS instead of Hilbert in the paper.
> - We conduct an ablation study for different SFCs in Sec. 5.4 and Table 4. Results indicate that the SIS/Z-Order achieves the best performance (80.438), and Hilbert is slightly worse (79.842), while Sweep and Scan will significantly damage the model performance.
>
> Q3: Typo issue
>
> A3: Thanks for pointing them out. We will carefully revise the manuscript again and correct these problems.
>
> [s12] Khadka, Shauharda, and Kagan Tumer. "Evolution-guided policy gradient in reinforcement learning." NIPS. 2018.
>
> [s13] Guo, Zichao, et al. "Single path one-shot neural architecture search with uniform sampling." ECCV. 2020.
>
> [s14] Yang, Zhaohui, et al. "Cars: Continuous evolution for efficient neural architecture search." CVPR. 2020.
>
> [s15] Jaderberg, Max, et al. "Population based training of neural networks." arXiv:1711.09846 (2017).
>
> [s16] Hassanat, Ahmad, et al. "Choosing mutation and crossover ratios for genetic algorithms—a review with a new dynamic approach." Information 10.12 (2019): 390.
>
> [s17] Dong, Yihe, Jean-Baptiste Cordonnier, and Andreas Loukas. "Attention is not all you need: Pure attention loses rank doubly exponentially with depth." arXiv:2103.03404 (2021).
>
> \* The paper index with **s** represents the paper we are going to add to the revised version.

---

### Official Review · Reviewer_TDFa · 2021-07-15

**Rating:** 6
**Confidence:** 4

**Summary:**

In this article, the author
1. explain the transformer from the EA algorithm perspective
2. uses the dynamic local population strategy used in EA algorithm and transfer the strategy into the transformer
3. introduces the spatial filling curve to transform the image data
4. achieves SOTA results on a number of tasks.

**Limitations And Societal Impact:**

Please see the main review part.

**Main Review:**

pros:
1. This article has a certain degree of innovation
2. The result of this article is good

cons:

Q1. The writing of this article is not clear

Q1.1 This article contains too much information, the author has too many things to say. But the main contribution is not described clearly. I suggest the author use a few sentences to describe the core contribution of this article and write it in the first part. List by points.

Q1.2 The framework of this method seems complex. It is suggested to insert a diagram on the second page and highlight the core contribution.

Q2. The necessity of introducing EA

Q2.1 What is the meaning of introducing EA algorithm? What are the unique advantages of local population strategy in EA algorithm? Why is it useful to introduce local information because EA is similar to transformer? In my opinion, the introduction of EA algorithm has some over packaging.

Q2.2 Again local? The purpose of introducing transformer into CV task is to deal with the image information directly in a uniform way, without considering the sequence relationship between the image patches, and deal with the problem of CV from a global perspective. This article seems to reuse some features of images, such as local information and sequence to transformer? This sounds like considering some special features of the image and used in the transformer. I don't think it's bad, but I think it might be contrary to the original intention of the transformer.

Q3. Method & Experiments

Q3.1 Please report the FLOPs.

Q3.2 Please add all the Efficientnet models in Fig. 6. It seems that the improvement is incremental.

Q3.3 About local. It actually introduces some priors of CNN's local information into transformer to a certain extent. The idea of local is not new and needs to quote more relevant papers.

Q3.4 About SFC. If the input image is not a regular rectangle, what is the impact? Is the strategy adopted by the author the optimal one for 2D image arrangement?

========== post rebuttal ============

I have read the rebuttal and the rebuttal addresses most of my concerns. I will change my score from 4 to 6.

The rebuttal should be merged into main paper in the next version.

**Time Spent Reviewing:**

4h

---

> ### Author Response · Authors · 2021-08-10
> **Rebuttal by Authors for Reviewer TDFa**
>
> Thanks to the reviewer for the comments, and we would like to clarify several things to address the reviewer's concerns:
>
> Q1: The writing of this article is not clear, and the main contribution is not described clearly
>
> A1: Thanks for the suggestion. Abstract, Introduction, and Method are organized according to our contributions, and all of the other three reviewers think our paper is easy to follow. We will highlight our core contributions in the revised version as follows:
> - In theory, we explain the rationality of Vision Transformer (TR) by analogy with Evolutionary Algorithm (EA) and derive that they have consistent mathematical representation.
> - For the method, we improve a unified EAT model by analogy with dynamic local population concept in EA and design a Task-related Head to deal with various tasks more flexibly and elegantly.
> - On framework, we introduce Space-Filling Curve (SFC) module as a bridge between rasterized 2D image data and serialized 1D sequence data. This makes it possible to integrate the unified paradigm that uses a unified model to solve multi-modal tasks, keeping the network architecture and data structure independent of the data dimensionality. Not that only several 1D operators are required for our method, meaning that it is very friendly to the underlying optimization workload of different platforms.
> - Massive experiments on classification and multi-modal tasks demonstrate the superiority and flexibility of our method.
>
> Also, we will enrich Figure 1 and Figure 2 to make our approach more understandable.
>
> Q2.1: The necessity of introducing EA
>
> A2.1: We do not see the introduction of EA as some packaging, as it does a good job of explaining TR structure and giving some insights into how to improve the network.
> - ***We hope to explain as much as possible why Transformer works well (in L125) while improving the model.*** Incidentally, we find that the operators in EA based on biological evolution are very similar to the operations in TR (i.e., Crossover vs. SA, Mutation vs. FFN, *etc*.), so we borrowed this proven practical heuristic algorithm to make a reasonable explanation for the structure of TR. Please refer to Reviewer NvRs-A1 for a more detailed explanation.
> - ***Proven improvements in EA have insight for improving Transformer.*** The concept of the dynamic local population had been successfully introduced to EA in some works [35, 37, 55], and it ensures better performance and stability. Also, there is a local relationship among the adjacent image patches in CV and the adjacent sequence tokens in NLP, so we analogously apply this idea to improve TR and extend a local path besides global MSA for a more efficient TR-based network.
> - ***We hope that everyone can be inspired by cross-disciplinary and interpretative research like EA vs. Transformer in this article.*** Also, as stated in Sec. 6, heuristic EA has been successfully applied to many subdomains, e.g., RL (PES [50], ERL[s12]), NAS (SPOS [s13], CARS [s14]), Hyperparameter Adjustment (PBT [s15]), *etc*. We will conduct a more rigorous exploration between heuristic EA and deep learning in the future.
>
> Q2.2: About the local information
>
> A2.2:
> - We agree that "The purpose of introducing transformer into CV task is to deal with the image information directly in a uniform way", but we disagree that ***"without considering the sequence relationship between the image patches".*** The naive transformer models the image input from a global perspective, but it also considers the sequence relationship implicitly for using positional embeddings (PE), e.g., learnable, absolute, or relative positional embeddings.
> - ***Introducing local information explicitly is not contrary to the intention of the vision transformer.*** Global TR and local CNN are not incompatible, and some recent works have combined them for better model effect, e.g., BotNet [s4], CPVT [16], CeiT [70], *etc*.
> - ***Our EAT is very different from existing fusion approaches that it is compatible for multi-modal tasks.*** Please refer to Reviewer NvRs-A2 for a more detailed explanation.
>
> Q3.1: About FLOPs
>
> A3.1: We explain the limitation of FLOPs metric and also evaluate the corresponding results.
>
> | **Method** | DeiT-Ti | EAT-Ti | DeiT-S | EAT-S | DeiT-B | EAT-B |
> |:-:|:-:|:-:|:-:|:-:|:-:|:-:|
> | **FLOPs (G)** | 1.257 | **1.017** (-19.1% $\downarrow$) | 4.605 | **3.827** (-16.9% $\downarrow$) | 17.577 | **14.828** (-15.6% $\downarrow$) |
>
> - FLOPs is not used by following DeiT's setting. As we all know, FLOPs does not reflect the real running speed well (e.g., EffNet-B3 is 1.8G that is much lower than DeiT-S's 4.6G, but its speed is even slower, 732.1 FPS vs. 940.4 FPS [36]); therefore, we report fairer throughput results for different methods in Table 1 and Sec. 5.2 in the paper.
> - We further evaluate FLOPs of our EAT-Ti/S/B in the above table, which reduce by a large margin (-19.1%$\downarrow$, -16.9%$\downarrow$, and -15.6%$\downarrow$, respectively) compared with the baseline DeiT-Ti/S/B. We will supplement the results in the final version.
>
> Q3.2: Add all the EfficientNet models in Fig. 6
>
> A3.2: Thanks for the suggestion. Like SwinTransformer [36], we choose only some EfficientNet models for comparison, which is enough to show EAT's advantages over Efficient, e.g., CPU throughput and inference memory. Admittedly, the improvement of EfficientNet is incremental on the GPU, and the EfficientNet in Fig. 6-1 outperforms the EAT model in some segments. However, strictly speaking, the comparison for EfficientNet here is unfair for our EAT. The EAT model is just trained at 224 resolution (while EfficientNet is from 224 to 600 along with B0 to B7), and more powerful models in Appendix E can easily overtake EfficientNet's effect. Also, the advantages over EfficientNet under a more robust setting have been fully illustrated in DeiT [56]. Last but not least, B1, B3, and B6 are not placed in Fig. 6 for the sake of the figure's space and aesthetics, and we will supplement these models as suggested.
>
> Q3.3: About local
>
> A3.3: Please refer to A2.2.
>
> Q3.4: About SFC
>
> A3.4: The SFC operation involved in the paper applies to the classification task that the image ratio is 1. For the irregular rectangle, SFC also applies because it is essentially a high-dimensional to one-dimensional mapping. More details can be seen in the wiki [https://en.wikipedia.org/wiki/Space-filling_curve] and [59]. Please refer to Reviewer Ppg5-A2 for a more detailed explanation.
>
> [s4] Srinivas, Aravind, et al. "Bottleneck transformers for visual recognition." CVPR. 2021.
>
> [s8] Dai, Zihang, et al. "CoAtNet: Marrying Convolution and Attention for All Data Sizes." arXiv:2106.04803 (2021).
>
> [s11] Li, Yawei, et al. "Localvit: Bringing locality to vision transformers." arXiv:2104.05707 (2021).
>
> [s12] Khadka, Shauharda, and Kagan Tumer. "Evolution-guided policy gradient in reinforcement learning." NIPS. 2018.
>
> [s13] Guo, Zichao, et al. "Single path one-shot neural architecture search with uniform sampling." ECCV. 2020.
>
> [s14] Yang, Zhaohui, et al. "Cars: Continuous evolution for efficient neural architecture search." CVPR. 2020.
>
> [s15] Jaderberg, Max, et al. "Population based training of neural networks." arXiv:1711.09846 (2017).
>
> \* The paper index with **s** represents the paper we are going to add to the revised version.

---

### Official Review · Reviewer_NvRs · 2021-07-15

**Rating:** 5
**Confidence:** 4

**Summary:**

The authors first explain the analogy between Transformer and the classic Evolutionary Algorithm (EA). Then, inspired by the dynamic local population in EA, the authors propose the EAT model, which integrates local operations (e.g., convolution, local attention, etc.) in the Transformer model. Besides, the authors design spatial-filling curve and task-related heads to improve the performance and generalization of the EAT model. Experiments on three tasks, including image classification, image retrieval and vision language navigation, demonstrate the effectiveness of the proposed EAT model.

**Limitations And Societal Impact:**

Overall, the contribution of this paper to the research community is not significant enough. The analogy between Transformer (TR) and Evolutionary Algorithm (EA) could not promote the explainability of TR. The integration of local and global representations is not novel in the literature of deep learning. The spatial-filling curve and the task-related heads do not show significant improvements in the experiments on ImageNet.

**Main Review:**

This paper explains the rationality of Vision Transformer by the analogy between Transformer and the classic Evolutionary Algorithm (EA). Based on this analogy, the authors improve the Transformer model by three modules, including the integration of local operations in the attention layer, spatial-filling curve, and the task-related heads.

First, the analogy between Transformer (TR) and Evolutionary Algorithm (EA) is not much convincing. Convolution, attention, and fully-connected layers are just common operations to integrate low-level representations to form high-level semantics. These three operations share the same mathematical representation of X’ = WX. I didn’t see that the analogy between TR and EA could extend the explainability of TR.

Second, as for the local path and global path in EAT, this design is essentially a way to fuse global and local representations. For a representation, some channels are used for global operations (i.e., MSA), other channels are used for local operations (i.e., CNN or local attention). However, the fusion of global and local information has been discussed in many papers like [1,2,3,4]. And the novelty in this paper is limited.

Third, the ablation studies in Table 4 shows the effectiveness of spatial-filling curve and task-related heads. However, the improvements of EAT over the state-of-the-art methods on ImageNet in Table 1 are not significant (e.g., EAT-B obtains 82.0 while DeiT-B obtains 81.8 under the same number of parameters).
The quality of paper writing is good. It is easy for readers to understand the ideas of this paper. All the figures and tables are well illustrated. And the whole paper is well organized.

[1] Wang, Xiaolong, et al. "Non-local neural networks." Proceedings of the IEEE conference on computer vision and pattern recognition. 2018.
[2] Cao, Yue, et al. "Gcnet: Non-local networks meet squeeze-excitation networks and beyond." Proceedings of the IEEE/CVF International Conference on Computer Vision Workshops. 2019.
[3] Qiu, Zhaofan, et al. "Learning spatio-temporal representation with local and global diffusion." Proceedings of the IEEE/CVF Conference on Computer Vision and Pattern Recognition. 2019.
[4] Srinivas, Aravind, et al. "Bottleneck transformers for visual recognition." Proceedings of the IEEE/CVF Conference on Computer Vision and Pattern Recognition. 2021.


**Time Spent Reviewing:**

8

---

> ### Author Response · Authors · 2021-08-10
> **Rebuttal by Authors for Reviewer NvRs**
>
> Thanks to the reviewer for the comments, and we would like to clarify several things to address the reviewer's concerns:
>
>
> Q1: Explainability of Transformer (TR) by analogy with Evolutionary Algorithm (EA) is not much convincing
>
> A1:
> - ***As explained in previous works [s5, s6], formula $X' = WX$ actually requires different mathematical assumptions for MLP, CNN, and ATT operations.*** This equivalent is generally unreasonable, where $W$ and $X$ in three operators have different concepts and calculation methods.
>     - For MLP, $W$ is regarded as a set of static parameters to enrich features for each point-wise $X$, which is the essential expression of $X' = WX$.
>     - For ATT, $W$ represents the weighted matrix obtained by dynamic calculation $softmax(\frac{QK^{\top}}{\sqrt{d_k}})$, which reorganizes the global feature $X$ and can be regarded as the extended-expression of $X' = WX$.
>     - For CNN, it can be seen as a locality-wise extension of point-wise MLP, no tautology here since no CNN operators in TR.
> - ***The operators in EA do a good job of explaining the sub-models in TR, reasons as follows.***
>     - As discussed above, ATT and MLP play different roles in the modeling of image patches.
>     - In global, ATT is responsible for the information interaction among all image patches, which has the same design idea as the global Crossover operator in EA. For mathematical consistency derivation, please refer to Eqs. 4-6 in the paper.
>     - In individual, FFN is used to extract the feature of each image patch, which is consistent with the Mutation operator for individual evolution in EA. For mathematical consistency derivation, please refer to Eqs. 7-9 in the paper.
>     - Furthermore, we find that the Residual Connection and Task-related Token modules also have analogous concepts in EA, i.e., Population Succession and Best Individual.
> - ***We will give more insightful explanations in three aspects: 1) necessity of module; 2) visual interpretability; and 3) application in other areas. Please refer to Reviewer Ppg5-A1 for details.***
>
> Q2: The novelty for global and local paths.
>
> A2: As we know, fusing global and local information is just a general idea to improve the network, but its specific implementation varies from paper to paper. Our fusion manner is obviously different from them in the following aspects:
> - The previous methods usually cascade (NL [s1], GCNet [s2], Xiao~*et al*. [s7], CoAtNet [s8]), parallel (LGD [s3], VOLO[s9], TNT [s10]), or replace (BotNet [s4], LocalViT [s11]) current layers with the new module for fusing global and local information, which is often accompanied by increases in parameters and calculation. Differently, we separate global MSA in channel dimension into global and local paths, capable of capturing better features while reducing the number of parameters (*c.f.* Proposition 1) and FLOPs (*c.f.* Proposition 2).
> - EAT's local idea is inspired by the concept of the dynamic local population in EA, which is enlightening rather than conceptual packaging. We hope this can inspire everyone to pay more attention to heuristic algorithms and cross-disciplinary inspiration for deep learning.
> - ***Our purpose is to design a unified model to handle multi-modal tasks simultaneously***. Thus our local design not only works for CV but also contributes to the local text of NLP. This local design is beneficial for our proposed unified paradigm.
>
> Q3: The improvements on ImageNet in Table 1 are not significant.
>
> A3:
>
> | **Method** | **Top-1** | **GPU Speed (FPS)** | **CPU Speed (FPS)** | **Inference Memory (M)** |
> |:-:|:-:|:-:|:-:|:-:|
> | DeiT-B | 81.8 | 290 | 10.2 | 2760 |
> | EAT-B | **82.0** (0.2 $\uparrow$) | **329** (13.4% $\uparrow$) | **11.7** (14.7% $\uparrow$) | **2508** (252 $\downarrow$) |
>
> - Our EAT is designed with the same amount of parameters as the baseline DeiT for fair comparison. Although EAT-B (82.0) only increases by +0.2 compared to DeiT-B (81.8), EAT has faster speeds than DeiT (329 vs. 290, **+13.4% $\uparrow$** for GPU and 11.7 vs. 10.2, **+14.7% $\uparrow$** for CPU) and a smaller inference memory occupancy. We consider EAT's balance of precision and efficiency to be significant, for the reason that the BoTNet (CVPR'21) improves only +0.2 (77.0, *c.f.* Table 9 in [s4]) over the weak baseline ResNet50 (76.8), but it runs **-31.3% $\downarrow$** slower and has **1.2 $\times \uparrow$** more FLOPs than the baseline.
> - Our method can steadily improve the accuracy and speed of the model, especially for the small network, e.g., improving +0.5 for the **columnar** DeiT-Ti in Table 1. We further apply our local idea to **pyramid** PVT-Ti[67] and consistently obtains +0.9 gains on Top-1. Also, more results under more robust settings and source code can be found in the supplementary material.
> - ***Our core contribution is to design a unified paradigm to handle multi-modal tasks, rather than simply achieving higher scores on the classification task***, as mentioned in the title and Introduction. We apply EAT to multi-modal tasks, and it obtains a significant performance improvement, e.g., +3.7 on CSS and +6.0 on Fashion200k for the TIR task, even though only using a unified network to handle both CV and NLP data.
>
> Also, we have again highlighted and listed our core contributions, and please refer to Reviewer TDFa-A1.
>
> [s5] Gong, Jingjing, et al. "Convolutional interaction network for natural language inference." EMNLP. 2018.
>
> [s6] Cordonnier, Jean-Baptiste, et al. "On the relationship between self-attention and convolutional layers." CVPR. 2020.
>
> [s7] Xiao, Wen, et al. "Extractive summarization of long documents by combining global and local context." EMNLP. 2019.
>
> [s8] Dai, Zihang, et al. "CoAtNet: Marrying Convolution and Attention for All Data Sizes." arXiv:2106.04803 (2021).
>
> [s9] Yuan, Li, et al. "Volo: Vision outlooker for visual recognition." arXiv:2106.13112 (2021).
>
> [s10] Han, Kai, et al. "Transformer in transformer." arXiv:2103.00112 (2021).
>
> [s11] Li, Yawei, et al. "Localvit: Bringing locality to vision transformers." arXiv:2104.05707 (2021).
>
> \* The paper index with **s** represents the paper we are going to add to the revised version.

---

### Decision · Program_Chairs · 2021-09-27

**Decision:**

Accept (Poster)

**Comment:**

This paper provides a view on the Transformer architecture as being related to that of an Evolutionary Algorithm. Based on this, the authors propose some improvements to the architecture and show convincing empirical results to demonstrate their efficacy and efficiency. The reviewers and discussion subsequent to the rebuttal support this view and I trust the authors will incorporate the reviewers' comments into the final manuscript.